# Genomic Insights and Antimicrobial Potential of Newly *Streptomyces cavourensis* Isolated from a Ramsar Wetland Ecosystem

**DOI:** 10.3390/microorganisms13030576

**Published:** 2025-03-03

**Authors:** Mabrouka Benhadj, Taha Menasria, Nawel Zaatout, Stéphane Ranque

**Affiliations:** 1Biomolecules and Application Laboratory, Faculty of Exact Sciences and Natural and Life Sciences, Echahid Cheikh Larbi Tebessi University, 12002 Tebessa, Algeria; mabrouka.benhadj@univ-tebessa.dz; 2Department of Applied Biology, Faculty of Exact Sciences and Natural and Life Sciences, Echahid Cheikh Larbi Tebessi University, 12002 Tebessa, Algeria; 3Department of Microbiology and Biochemistry, Faculty of Natural and Life Sciences, University of Batna 2, 05078 Batna, Algeria; t.menasria@univ-batna2.dz (T.M.); n.zaatout@univ-batna2.dz (N.Z.); 4Aix Marseille University, SSA, RITMES, 13005 Marseille, France; 5IHU-Méditerranée Infection, 19-21 Boulevard Jean Moulin, 13005 Marseille, France

**Keywords:** wetland, *Streptomyces cavourensis*, antimicrobial activity, whole genome sequencing, bioactive molecules

## Abstract

The growing threat of antimicrobial resistance underscores the urgent need to identify new bioactive compounds. In this study, a *Streptomyces* strain, ACT158, was isolated from a Ramsar wetland ecosystem and found to exhibit broad-spectrum effects against Gram-positive and Gram-negative bacteria, as well as fungal pathogens. The active strain was characterized as *S. cavourensis* according to its morphology, phylogenetic analysis, average nucleotide identity (ANI), and digital DNA–DNA hybridization (dDDH). Whole-genome sequencing (WGS) and annotation revealed a genome size of 6.86 Mb with 5122 coding sequences linked to carbohydrate metabolism, secondary metabolite biosynthesis, and stress responses. Genome mining through antiSMASH revealed 32 biosynthetic gene clusters (BGCs), including those encoding polyketides, nonribosomal peptides, and terpenes, many of which showed low similarity to known clusters. Comparative genomic analysis, showing high genomic synteny with closely related strains. Unique genomic features of ACT158 included additional BGCs and distinct genes associated with biosynthesis pathways and stress adaptation. These findings highlight the strain’s potential as a rich source of bioactive compounds and provide insights into its genomic basis for antimicrobial production and its ecological and biotechnological significance.

## 1. Introduction

Microorganisms have long been harnessed for the production of natural products, particularly due to their ability to synthesize secondary metabolites with significant bioactivity [1]. Among these, actinomycetes, a diverse group of Gram-positive bacteria, are extensively recovered in both natural and manmade environments [2]. They are the key contributors to the biosynthesis of novel and unique secondary metabolites with diverse structures and functions, including antibacterial, antifungal, antitumor, antiprotozoal, and antiviral activities, as well as the production of vitamins and hydrolytic enzymes [3].

The genus *Streptomyces* is a prominent representative of actinomycetes, characterized by aerobic, Gram-positive, non-acid-fast, and multicellular bacteria that form extensively branched substrate and aerial mycelia [4]. Upon maturation, the aerial mycelia differentiate into chains of multiple spores. *Streptomyces* are ubiquitous in terrestrial ecosystems and are widely recognized for their remarkable metabolic versatility and adaptability [5]. They play a crucial role in the degradation of organic matter, such as leaf litter, thereby contributing significantly to nutrient cycling and soil fertility [6]. Moreover, *Streptomyces* can metabolize a wide range of complex compounds, including polysaccharides, alcohols, amino acids, and aromatic compounds, through the production of extracellular enzymes such as amylase, chitinase, cellulase, glucanase, and protease [7]. Notably, *Streptomyces* is recognized as the largest antibiotic-producing genus identified in the microbial world to date [8]. In addition to antibiotics, it has the ability to produce other important bioactive secondary metabolites, including antifungals, antivirals, antitumor agents, antihypertensives, and notably, immunosuppressants [3]. As of September 2024, 1228 species and 74 subspecies with 773 validly published and correct names had been approved, as in the List of Prokaryotic names with Standing in Nomenclature [9]. This number continues to increase annually with several dozen new species being proposed each year.

The recent rise of antibiotic resistance, particularly from carbapenem-resistant bacteria, has become a critical global health concern, complicating the treatment of bacterial infections [10]. These organisms, including strains of *Klebsiella pneumoniae*, *Escherichia coli*, and *Acinetobacter baumannii*, exhibit resistance to carbapenems, which are often considered last-resort antibiotics for multidrug-resistant infections [11]. This growing resistance reduces the effectiveness of existing antibiotics, leading to prolonged illnesses and increased mortality rates. Consequently, this alarming trend underscores the urgent need for research into alternative sources of potent bioactive compounds, especially from actinomycetes [12]. Exploring these sources may yield new antibiotics that can effectively combat resistant strains and enhance the therapeutic arsenal against challenging infections [8].

Significant advancements in DNA sequencing technologies have led to a substantial increase in the number of fully sequenced actinomycetes genomes [13]. As a result, a panel of bioinformatics tools for genome annotation and mining have been developed to identify secondary metabolite biosynthetic gene clusters (smBGCs) and predict their location [14]. Recent in silico large-scale studies have highlighted the tremendous potential of *Streptomyces* species for discovering novel and potentially relevant compounds associated with BGCs, many of which remain uncharacterized and underscore the significant untapped biosynthetic potential of *Streptomyces*, particularly those from poorly studied ecological niches [15].

Algeria hosts an exceptional array of wetlands and hypersaline lakes, recognized globally for their unique typology and ecology. Among these, 50 sites are classified as internationally significant Ramsar sites [16]. These ecosystems span diverse climatic zones, from subtropical wetlands in the coastal northeast to semi-arid regions in the Hauts Plateaux and arid regions of the Sahara, fulfilling essential ecological functions [17]. Despite their ecological importance, the microbial diversity and bioactivity within these habitats remain poorly understood. Thus, exploring these unique ecosystems not only enhances our understanding of microbial diversity but also opens avenues for discovering new compounds that can contribute to drug development and ecological sustainability. In this context, we studied the genome of a selected antimicrobial-producing *Streptomyces* strain, isolated from a natural representative Ramsar wetland in the Mediterranean region (Fetzara Lake, northeastern Algeria). This study combines genomic, metabolic, and physiological analyses to explore the strain’s antimicrobial potential, carbohydrate-active enzymes, and biosynthetic gene clusters (BGCs).

## 2. Materials and Methods

### 2.1. Actinomycete Isolation

The ACT158 strain was isolated from Fetzara Lake, using starch casein medium as described by Benhadj et al. [18] supplemented with 2.5 μg/mL of rifampicin, 10 μg/mL of amphotericin B, and 75 μg/mL of fluconazole. The plates were incubated at 28 °C for 10 days and the strain was subcultured and maintained on ISP2 agar at 4 °C and at −80 °C on the 20% of glycerol as mycelia suspension.

### 2.2. Phenotypic and Physiological Characterization

The strain S158 was characterized based on its morphological, physiological, and biochemical traits using standard methods [19,20]. Morphological characterization was conducted according to the International *Streptomyces* Project (ISP), employing ISP1, ISP2, ISP3, ISP4, ISP5, and ISP6 media. Growth characteristics were observed after 3, 7, 14, and 21 days of cultivation at 30 °C. ISP6 and ISP7 were utilized to assess melanoid pigment production. NaCl tolerance, pH, and temperature growth were tested on ISP2 medium. Nitrate reduction, hemolytic activity, and the production of amylase, gelatinase, coagulase, urease, citrate permease, and esterase were analyzed using various media as described by Benhadj et al. [18]. The minimal basal medium (ISP9) was used to determine the capacity of the isolates to use different carbon sources at 1% of final concentration and A range of sugars, including arabinose, ribose, rhamnose, mannitol, fructose, sucrose, galactose, mannose, inositol, glucose, and maltose, were tested for acid production.

### 2.3. MALDI-TOF MS Identification

To characterize ACT158, protein extraction was performed using the ethanol/formic acid (EtOH/FA) protein extraction method as described by Cassagne et al. [21]. Subsequently, MALDI-TOF MS analysis was conducted using a Microflex MALDI-TOF mass spectrometer (Bruker Daltonics, Bremen, Germany). Extracted samples were prepared for analysis by spotting them onto polished MALDI plates and overlaying them with a matrix solution of α-cyano-4-hydroxycinnamic acid (HCCA). Calibration of the spectrometer was performed using *Escherichia coli* DH5α, and a non-inoculated matrix served as a negative control. To ensure data reliability, the sample was analyzed in triplicate and the identification confidence was evaluated based on score thresholds, with scores ≥1.9 considered low confidence and scores <1.7 deemed unreliable. Normalized MALDI-TOF MS spectra of ACT158 and reference strains were subjected to agglomerative hierarchical clustering (AHC) analysis to assess the phylogenetic relationship of ACT158 within the *Streptomyces* genus.

### 2.4. DNA Extraction, 16SrRNA Gene Sequencing and Phylogenetic Analysis

The genomic DNA of the ACT158 strain was extracted as described by Kieser et al. [22]. The 16S rRNA gene was amplified by PCR using a pair of the universal primers Fd1 (5′ AGAGTTTGATCCTGGCTCAG) and rP2 (5′-AAGGAGGTGATCCAGCC) [23]. Amplified product was sequenced and homology searches were conducted by comparing sequences with those in public databases (NCBI) using the Basic Local Alignment Search Tool (BLAST), v. 2.16.0 and EzBioCloud database (http://www.ezbiocloud.net/, accessed on 10 January 2025) [24]. Phylogenetic analyses were performed using MEGA software version 11, with sequences aligned against neighboring nucleotide sequences using CLUSTALW [25]. The phylogenetic tree was constructed using the neighbor-joining (NJ) method, and the topologies were evaluated through bootstrap sampling based on 1000 replicates.

### 2.5. Antimicrobial Testing Activity of Streptomyces sp. ACT158

The antibacterial activity of *Streptomyces* sp. ACT158 was evaluated using the agar diffusion method as described by Benhadj et al. [8]. Mycelium plugs (7 mm) deposited onto LB (Luria and Bertani) plates inoculated with bacterial indicator strains (Appendix A) at an optical density of 0.04 and the activity was assessed over 3, 7, 10, 14, 17, and 21 days. Two culture media, ISP2 and glycerol yeast extract agar (GYEA), were used to optimize bioactive molecule production. Similarly, the antifungal activity of ACT158 was screened using the agar plug and double-layer methods. In the latter, ACT158 was incubated on ISP2 plates for 10 days, followed by overlaying with ISP2 agar inoculated with the target organism. Inhibition zones were measured after 48 h and 10 days at 30 °C for yeast and molds, respectively.

### 2.6. Genome Sequencing, Assembly, Gene Prediction and Functional Annotation

ACT158 was cultured in ISP2 solid medium and incubated at 28 °C for 4 days. After incubation, DNA was extracted as described previously [18]. The genomic DNA of the ACT158 strain was fragmented to 150 bp, and libraries were constructed and sequenced using by paired-end strategy with Illumina Mi-seq technology. The raw data were quality-controlled using the FastQC v. 0.11.9 [26], and reads were used assembled de novo using the hybrid spades program (SPAdes v. 3.13.0) [27].

The genome of *Streptomyces* sp. ACT158 was comprehensively annotated using multiple bioinformatics tools and databases. Gene prediction and functional annotation were encoded using the NCBI Prokaryotic Genome Annotation Pipeline [28] and rapid annotations using subsystems technology (RAST) (http://rast.nmpdr.org/, accessed on 10 December 2024) [29,30]. Orthologous genes were annotated using EggNOG mapper (v.2.1.12) to infer functional and structural insights. Additionally, metabolic pathways were reconstructed using GhostKOALA, integrated with KEGG (Kyoto Encyclopaedia of Genes and Genomes) pathway data [31]. The proteomes of *Streptomyces* sp. ACT158 was subjected to automated annotation and assignment to CAZymes [32] using the dbCAN3 meta server and HMMdb v12.0, identified by at least two tools within the server including HMMER, dbCAN, DIAMOND, CAZy, and dbCAN-sub. The biosynthetic potential of the ACT158 was explored through antiSMASH (v.7) for functional annotation and mining of natural products’ biosynthetic gene clusters [33]. The graphical representation of the genome was generated using Proksee [34].

### 2.7. Phylogenomics, and Comparative Genome Analysis

The genome sequence data were uploaded to the Type (Strain) Genome Server (TYGS) (https://tygs.dsmz.de, accessed on 10 December 2024), for a whole genome-based taxonomic analysis [35]. The resulting intergenomic distances were used to infer a balanced tree with branch support via FASTME 2.1.6.1 including SPR postprocessing and branch support was inferred from 100 pseudo-bootstrap replicates each. The Genome-to-Genome Distance Calculator (GGDC) was employed for species delineation (http://ggdc.dsmz.de/home.php, accessed on 10 December 2024) [36], and the estimate DDH resulted from a generalized linear model (GLM). The recommended cutoff value for species delimitation using GGDC was 70% similarity. Additionally, average nucleotide identity (ANI) values were calculated using the BLAST-based OrthoANI tool [37]. Comparative genome analysis of COGs in *Streptomyces* sp. ACT158 and the most closely relative species including *S. cavourensis* DSM41795 (GCA_006788935.1), *S. cavourensis* JCM4298 (GCA_014649215.1), *S. rhizosphaericola* 1AS2c (GCA_004794175.1), and *S. nanshensis* SCSIO10429 (GCA_001751255.1) were performed using a BLAST-based analysis with OrthoVenn v.3 [38] under default parameters. Subset genes, such as species group shared and unique subsets of genes within individual genomes, were identified by clustering the results from the core and pan-genome calculations.

## 3. Results

### 3.1. Phenotypical Characterization of ACT158

The micro-morphological features of *Streptomyces* sp. ACT158 were observed after cultivation on ISP2 medium at 30 °C. The strain is a Gram-positive bacterium characterized by long filaments capable of producing aerial mycelia and spores. The strain exhibited good growth on ISP1, ISP2, ISP4, and ISP6 media, moderate growth on ISP3 and ISP7, and displayed various mycelium colors, including pale yellow, beige, and brown. Diffusible melanoid pigments were detected on all tested media except ISP3 and ISP4 (Appendix A). The strain grew well at 40 °C but not at 4 °C, with an optimal growth temperature of 30 °C at a salt concentration of up to 5%, and could grow across a pH range (3–10). *Streptomyces* sp. ACT158 presents functional diversity, producing enzymes such as amylase, coagulase, gelatinase, citrate permease, nitrate reductase, cellulase, caseinase, and esterase. Hemolytic and catalase activities were also observed. The strain utilized mannitol, glucose, sorbitol, sucrose, and melibiose as carbon sources, with accompanying acid production. However, it could not metabolize arabinose, rhamnose, or inositol.

### 3.2. MALDITOF and 16SrRNA-Based Analysis

The strain was subjected to identification using MALDITOF-MS and raw mass spectrometry profiles were clustered to validate the accuracy of the MALDI Biotyper identification. The results showed that ACT158 produced reliable results affiliated with *Streptomyces cavourensis* with a log score of 2.29 ± 0.02. Representative spectra profiles of ACT158 are shown in Appendix A. The phylogenetic analysis of the ACT158 strain was conducted based on 16S rRNA gene sequencing. Using the EzBioCloud database, nearly the complete sequence of the 16S rDNA gene (1487 bp) was identified. The sequence analysis revealed a 100% similarity with the type strain of *S. cavourensis* NBRC 13026^T^ and 99.38% with *S. bacillaris* NBRC 13487^T^ (Figure 1).

### 3.3. Genome Assembly, Annotation and Functional Analysis

Genome sequence assembly and general genomic features of ACT158 are presented in Table 1 and Figure 2. The circular chromosome was found to be 6,858,072 bp with a GC content of 71.1%. The annotated genome included 5122 protein-coding genes (CDSs), with a coding ratio of 71.1% and 639 putative pseudogenes. The chromosome included 3 rRNAs, 54 transfer RNAs genes, and 8 clustered regularly interspaced short palindromic repeats-CRISPR-associated (CRISPR-Cas) system.

A comprehensive genomic analysis of *Streptomyces* sp. ACT158 reveals its extensive metabolic capabilities and functional diversity. Based on KEGG pathway analysis (Appendix A), 2410 genes, or 35.3% of all CDSs, were mapped to 41 KEGG pathways highlighting the strain’s involvement in key metabolic, including amino acid metabolism, carbohydrate metabolism, secondary metabolite biosynthesis, energy production, nucleotide metabolism, and environmental information processing. These pathways underline its metabolic adaptability and potential for bioactive compound production.

In addition, COG (Clusters of Orthologous Groups) functional classification provided insights into gene clusters associated with 23 functional groups based on the COG codification (Appendix A). Major contributions included genes related to A (RNA processing and modification), B (chromatin structure and dynamics), C (energy production and conversion), D (cell cycle control and mitosis), and E (amino acid transport and metabolism), with the latter showing the highest representation (966 genes). Other significant categories included carbohydrate transport and metabolism (687 genes), transcription (987 genes), and energy production and conversion (617 genes). Additional classifications, such as cell wall/membrane biogenesis (285 genes), signal transduction, and secondary metabolite biosynthesis, further highlight the strain’s capacity for complex metabolic and regulatory processes. Rapid Annotations using Subsystems Technology (RAST) offered further details on functional diversity, with 275 subsystems for carbohydrate metabolism, more than 410 subsystems for protein, amino acid, and derivative metabolism, and 44 subsystems related to membrane transport (Appendix A). Stress response genes and pathways for DNA repair and secondary metabolite production were also identified, showcasing its ecological resilience and industrial potential. About 196 carbohydrate-active enzymes (CAZymes) were annotated (Appendix A), including 90 glycoside hydrolases (GHs) such as amylases (GH13, 15), chitinase (GH18, 19, 23), chitosanase (GH5), and cellulase (GH6), along with 39 glycosyl transferases (GTs), 28 auxiliary activity enzymes (AAs), and 22 carbohydrate-binding modules (CBMs). This diversity indicates the strain’s ability to degrade complex carbohydrates and its role in polysaccharide breakdown, which is crucial for natural product biosynthesis and biotechnological applications.

### 3.4. Similarity of Whole Genomes

The phylogenomic analysis of the ACT158 strain was conducted using genome sequences from related *Streptomyces* taxa, as shown in Figure 3. The phylogenomic tree, inferred with FastME based on GBDP distances, revealed the clustering of ACT158 with *Streptomyces cavourensis* strains JCM 4298 and DSM 41796, supported by high bootstrap values (>60%) and an average branch support of 89.1% (Figure 3). Genomic similarity assessments were evaluated using average nucleotide identity (ANI) and in silico DNA–DNA hybridization (DDH), with the results summarized in Figure 3 and Figure 4. ANI analysis revealed a 99.2% similarity between ACT158 and *S. cavourensis* DSM 41795^T^ and *S. cavourensis* JCM 4298^T^ (Figure 5). Similarly, DDH analysis revealed 90.4% and 90.3% similarity between ACT158 and both reference strains. These values exceed the species delineation thresholds of ≥95% for ANI and ≥70% for in silico DDH, confirming the taxonomic identification of ACT158. The heat map of the similarity matrix further supported these findings, clearly differentiating ACT158 from other *Streptomyces* species and reinforcing its identification as a strain of *S. cavourensis*.

### 3.5. Comparative Genome Analysis

Comparative genomic analysis of ACT158 and four *Streptomyces*-type strains revealed substantial gene family conservation (Figure 5). A significant proportion, 73.03% (4955 out of 6784), of identified gene families were shared among all five species. Among these strains, *S. cavouensis* DSM 41795 and *S. cavourensis* JCM4298 exhibited the highest number of gene families (6384 and 6849, respectively), whereas *Streptomyces* sp. ACT158 presented 5792 gene families. Notably, a high degree of gene family conservation was observed within the S. cavourensis species complex, with 90.8% (5774 out of 6361) of identified gene families (Figure 5). Gene Ontology (GO) enrichment analysis of significantly enriched, commonly shared gene families across all strains revealed associations with fundamental biological processes, including metabolic processes (GO:0008152), biological processes (GO:0008150), cellular metabolic processes (GO:0044237), and nitrogen compound metabolic processes (GO:0006807) (Appendix A).

### 3.6. Antimicrobial Activity of ACT158

The antimicrobial activity of *Streptomyces* sp. ACT158 was evaluated against a panel of bacterial and fungal strains (Figure 6). The strain exhibited notable antibacterial activity against all tested bacterial indicators except *Pseudomonas aeruginosa* clinical isolates. The inhibition zones varied depending on the incubation period, indicator strain, and production medium. On GYEA medium, the highest activity against bacteria was observed on day 3, with *Staphylococcus aureus* exhibiting a 25 mm inhibition zone. By day 7, the activity was more pronounced against *Salmonella* Typhimurium ATCC 14028 in ISP2 medium, measuring 19 mm. Overall, *Streptomyces* sp. ACT158 exhibited potent antagonistic activity against both Gram-positive and Gram-negative bacteria, with GYEA appearing as the most suitable medium for producing antibacterial bioactive molecules.

The results of antifungal screening indicated inhibition of all tested yeasts except Candida parapsilosis LCP, with inhibition zones ranging from 11.5 mm to 36 mm in diameter. For filamentous fungi, the antifungal activity was particularly notable. The strain displayed high inhibition against *Penicillium chrysogenum* (43.5 mm) (Figure 7), *Paecilomyces variotii* (25.5–39.5 mm), and *Rhizopus oryzae* (26–42 mm). Antifungal activity started as early as day 3, with pronounced effects against *Penicillium* and other filamentous fungi. Both ISP2 and GYEA media supported antifungal metabolite production, although GYEA was slightly more suitable. The graphical representation further confirms the medium-dependent variation in antimicrobial activity across tested strains and incubation times. These findings highlight the versatility of *Streptomyces* sp. ACT158 in producing bioactive compounds with broad-spectrum antimicrobial properties, underscoring its potential for biotechnological and pharmaceutical applications.

### 3.7. Analysis of Secondary Metabolic Biosynthetic Gene Clusters

The genome of *Streptomyces* sp. ACT158 was analyzed using the antiSMASH platform, revealing the presence of 32 predicted secondary metabolic biosynthetic gene clusters (BGCs) (Table 2). These clusters included ribosomally synthesized and post-translationally modified peptides (RiPPs), lanthipeptide class II, III, nonribosomal peptide synthetases (NRPSs), siderophores, butyrolactones, ectoine, and terpenes. Several of these BGCs demonstrated high similarity to known biosynthetic gene clusters, suggesting their potential role in producing bioactive compounds. Among the identified BGCs, noteworthy clusters included an NRPS-T1PKS cluster with 100% similarity to the naringenin biosynthetic gene cluster (BGC0001310) and an NRPS-T3PKS cluster identical to alkylresorcinol (BGC0000282). Additionally, the NRPS clusters exhibited varying similarities to well-characterized compounds such as jadomycin (42%), valinomycin/montanastatin (73%), and ectoine (75%) (Figure 8). Other significant hits included siderophore desferrioxamine B with 100% similarity and terpene geosmine, also with 100% similarity. Interestingly, the analysis also identified BGCs with lower similarity but linked to important bioactive molecules. These included an NRPS cluster with 50% similarity to the heat-stable antifungal factor (BGC0002365), a butyrolactone cluster with 29% similarity to showdomycin (BGC0001778), and a RiPP cluster with 22% similarity to lactazole (BGC0000606). Additionally, clusters for antibiotics like kinamycin (13%), steffimycin D (16%), and bafilomycin B1 (27%) were detected, suggesting that ACT158 may encode novel derivatives of these potent compounds, further underscores the strain’s capability to produce complex secondary metabolites (Figure 8).

## 4. Discussion

Actinomycetes, particularly *Streptomyces* species, are well-documented for their ability to produce a vast array of antibiotics and other bioactive compounds [39]. These microorganisms, found in diverse ecological niches, have evolved to produce an extensive array of secondary metabolites that are of considerable interest in the pharmaceutical, agricultural, and industrial sectors [12]. This metabolic diversity is closely tied to their ecological role in nutrient cycling and organic matter degradation [40,41]. Wetlands, often considered one of the most biodiverse ecosystems, are home to a range of microorganisms that play critical roles in biogeochemical cycles [17]. The environmental conditions in these ecosystems, such as fluctuating oxygen availability, nutrient concentration, and other factors, impose selective pressures on the microbial community [18]. This dynamic environment can lead to the emergence of microorganisms capable of producing a wide range of bioactive secondary metabolites, including enzymes and antimicrobial compounds [16].

In this study, a strain of *Streptomyces* sp. ACT158, isolated from such a unique and typical environment (Fetzara Lake), could unlock specific biochemical pathways that enable the strain to thrive in these conditions. Phenotypic characterization revealed ACT158’s ability to grow on multiple media types, including ISP1, ISP2, ISP4, and ISP6, while producing aerial mycelium and diffusible pigments. The pigmentation in actinomycetes is a hallmark of stress response pathways often linked to secondary metabolite biosynthesis [42]. Additionally, the strain’s capacity to produce extracellular enzymes such as amylase, cellulase, and protease highlights its functional versatility and genomic organization [43]. These enzymes not only facilitate nutrient acquisition in diverse environments but also provide precursors for secondary metabolite synthesis [44,45].

Such metabolic plasticity enhances the strain’s adaptability, a crucial trait for industrial large-scale processes requiring stable microbial performance under variable conditions [46]. Cellulases are widely used in the biofuel industry for the conversion of lignocellulosic biomass into fermentable sugars, a crucial step in the production of bioethanol [47]. In the food industry, amylases are employed for starch hydrolysis, which is fundamental in brewing, baking, and other food production processes [48]. Similarly, proteases are crucial in detergent manufacturing, food processing, the leather industry, pharmaceutics, and waste management [49].

The phylogenomic analysis of the ACT158 strain confirms its classification within the *S. cavourensis* species. Phylogenetic reconstruction using genome sequences from related taxa revealed a close relationship between ACT158 and *S. cavourensis*. Genomic similarity assessments using ANI and dDDH further corroborated this identification. *S. cavourensis* has been previously identified in a wide range of ecological niches, including marine ecosystems, agricultural soils, and plants [50,51]. This broad habitat distribution highlights its exceptional adaptability to diverse environmental conditions and its ecological role in nutrient recycling and organic matter degradation. To further understand the genomic characteristics of *Streptomyces* sp. ACT158, it is essential to consider the importance of genomic relatedness indices in species classification. Modern methods such as Average Amino Acid Identity (AAI), Digital DNA-DNA Hybridization (dDDH), and Average Nucleotide Identity (ANI) have become the gold standard for accurately determining the species and resolving the taxonomic status of complex taxon [52], particularly within the genus *Streptomyces* [53]. These methods provide a reliable framework for distinguishing between species and identifying new strains with unique properties in cases where traditional phenotypic methods may not provide sufficient resolution for species identification [54].

The genome of ACT158, with a size of approximately 6.8 Mb and a GC content of 71.1%, is typical of the *Streptomyces* genus, which is notable among bacteria for their large linear chromosomes with terminal inverted repeats and range in size from 6 to 12 Mb, encoding between 5300 and 11,000 proteins [55]. The genome annotation revealed a wealth of protein-coding genes associated with amino acid metabolism, carbohydrate transport, energy production, secondary metabolite biosynthesis, and other biologically relevant compounds. Functional annotation of the genome provides insights into the roles of individual genes and their contributions to the overall metabolic network of the organism [56]. This process involves assigning putative functions to genes based on sequence similarity to entries in databases like KEGG, dbcan, and COG [57].

Enzymes are fundamental to metabolic processes, enabling organisms to adapt to diverse environmental conditions and produce valuable secondary metabolites [58]. The ACT158 genome typically contains a rich repertoire of genes encoding enzymes involved in primary and secondary metabolism. The prediction of genes encoding extracellular hydrolytic enzymes, such as cellulases, amylases, and proteases, corroborates the strain’s enzymatic activities observed in phenotypic assays. For example, the prediction of amylase, kitinase, and cellulase involved in carbohydrate degradation, along with the identification of proteases that play critical roles in nutrient acquisition and stress response, is of interest for industrial applications like biofuel production and bioremediation and other applications [48,49].

In addition, the genome of *Streptomyces* sp. ACT158 may encode lipases and esterases, which are widely used in industries ranging from food and detergents to pharmaceuticals [59]. The presence of genes encoding oxidoreductases, such as peroxidases, is particularly noteworthy, as these enzymes are essential for the degradation of complex organic and aromatic compounds [60]. This capability makes *Streptomyces* sp. ACT158 a strong candidate for applications in environmental biotechnology, such as waste management and pollutant degradation.

A key outcome of genomic annotation is the identification of genes associated with stress response and adaptation to environmental conditions [61]. For *Streptomyces* sp. ACT158, genes encoding heat-shock proteins and oxidative stress enzymes are identified, reflecting the strain’s ability to survive in the dynamic and challenging conditions of wetlands. Genome analysis of the endophyte *Streptomyces* sp. KLBMP 5084 has highlighted their capacity to adapt to environmental stresses by encoding superoxide dismutases, peroxidases, catalases, and proteins involved in heavy metal resistance, as well as cold and heat shock proteins [62]. Similarly, a genomic investigation of *Streptomyces* sp. H-KF8 identified numerous genetic determinants associated with heavy metal resistance (49 genes), oxidative stress (69 genes), and antibiotic resistance (97 genes) [63]. Additionally, genes related to heavy metal resistance and osmotic stress suggest a capacity for bioremediation in contaminated environments. The annotation also uncovers genes involved in regulation and cell signaling networks, which play a crucial role in secondary metabolite production and environmental adaptation [61]. Functional analysis reveals a well-developed primary metabolism that supports the biosynthesis of precursors for secondary metabolites: amino acid derivatives, glycolysis, TCA cycle, and the pentose phosphate pathways provide energy and biosynthetic intermediates required for the production of polyketides and nonribosomal peptides [64].

The identification of enzyme systems involved in xenobiotic degradation further underscores the strain’s environmental adaptability. These systems often include monooxygenases, dioxygenases, and hydrolases that catalyze the breakdown of toxic compounds [65]. These predictions align with the ecological niche of *Streptomyces* sp. ACT158, suggesting that the strain has adapted to thrive in wetlands, which are often rich in organic matter and potential pollutants and may reveal enzymes with unexplored catalytic activities, opening new avenues for biotechnological innovation. Experimental validation of predicted enzymes is crucial to confirm their functions and assess their industrial potential. For *Streptomyces* sp. ACT158, validation studies can focus on key enzymes identified in biosynthetic gene clusters (BGCs) or those implicated in the degradation of complex polymers. The integration of enzyme prediction with other omics approaches, such as transcriptomics and proteomics, enhances the understanding of enzyme functionality under different conditions [66]. For example, transcriptomic data can reveal which enzyme-encoding genes are actively expressed in response to specific environmental stimuli, such as nutrient availability or stress factors [67].

The antimicrobial potential of *Streptomyces* sp. ACT158 further enhances its biotechnological appeal. The strain was found to exhibit broad-spectrum antimicrobial activity against both Gram-positive and Gram-negative bacteria (*Staphylococcus aureus*, *Bacillus subtilis*, *Micrococcus luteus*, *Klebseilla pneumoniae,* and *Salmonella* Typhimurium), as well as fungal pathogens such as *Aspergillus* and *Candida*. Historically, *Streptomyces* species have been a valuable source of antibiotics, producing around 100,000 compounds, which account for 70–80% of all natural bioactive products with pharmacological or agrochemical such as streptomycin, tetracycline, and erythromycin [3]. The broad-spectrum antimicrobial activity displayed by *Streptomyces* sp. ACT158 suggests the production of different antimicrobial agents. *Streptomyces* produce a variety of natural products with high structural diversity, including macrolides, tetracyclines, aminoglycosides, glycopeptides, ansamycins, and terpenes [68]. However, the continued rise in antimicrobial resistance globally calls for the discovery of new compounds, as existing antibiotics may become less effective. With evolving resistance mechanisms in pathogens, exploring new strains like *Streptomyces* sp. ACT158 offers great promise in developing innovative solutions to combat resistant infections [69,70].

Antimicrobial activity was particularly pronounced in GYEA compared to ISP2 media, where inhibition zones against key pathogens reached up to 25 mm. The strain also demonstrated potent antifungal properties, effectively suppressing the growth of yeast and filamentous fungi such as *Candida* spp., *Aspergillus* spp., *Penicillium* spp., *Paecilomyces* spp., and *Rhizopus*. The rapid onset of antifungal activity, observed within three days, suggests a highly efficient secondary metabolite production mechanism. The production of antimicrobial compounds by *Streptomyces* strains is strongly influenced by culture conditions, including media composition, incubation time, and the organisms used in testing [71]. Carbon sources, in particular, play a pivotal role in regulating antibiotic synthesis. Research has shown that preferred carbon sources can suppress secondary metabolism, which may reduce antibiotic production [72]. This regulation could explain the varying inhibition patterns observed in *Streptomyces* sp. ACT158 antagonistic assays when tested in different media, highlighting the importance of tailoring environmental conditions to enhance their antimicrobial activity. The strain *S. cavourensis* NRRL 2740 has been identified as a producer of several bioactive compounds, including the antitumor agent chromomycin [73], the antifungal metabolite flavensomycin, and humidin [74]. Additionally, macrolide antibiotics such as bafilomycins (B1, C1, and D) and hygrobafilomycin, known for their potent antifungal properties, have been associated with *S. cavourensis* species [75].

Genomic analysis provided further insights into the strain’s biosynthetic potential. AntiSMASH analysis predicted secondary metabolic biosynthetic gene clusters (BGCs), including those for polyketide synthases (PKS), nonribosomal peptide synthetases (NRPS), terpenoids, and other classes of bioactive compounds. The presence of multiple BGCs suggests that the strain is capable of producing a wide variety of bioactive compounds, which could be of significant interest for pharmaceutical and industrial applications. In addition, the presence of BGCs with low similarity to known clusters can uncover novel antibiotics and other bioactive molecules with therapeutic potential.

Polyketides are a diverse class of secondary metabolites with notable pharmacological activities, including antibiotic, anticancer, and immunosuppressive properties [76]. Notably, one NRPS-T1PKS cluster displayed complete similarity to the biosynthetic gene cluster (BGC) of the flavonoid naringenin, while another NRPS-T3PKS cluster was associated with alkylresorcinol. Although naringenin is primarily known as a plant-derived metabolite, it has recently been identified in a few *Streptomyces* species [77]. Furthermore, its encoding gene cluster has also been detected in certain *Saccharotrix* and *Kitasatospora* species [78]. Several biological activities have been associated with these compounds, including antioxidant, antitumor, antiviral, antibacterial, antifungal, and anti-inflammatory [79]. Additionally, Type II PKS clusters, associated with aromatic polyketides like jadomycin were detected. These clusters employ an iterative mechanism of chain elongation and cyclization, producing highly conjugated and planar structures typical of compounds with antibacterial and anticancer properties [80]. The identification of Type III PKS clusters, which catalyze the condensation of acyl-CoA precursors without requiring a carrier protein, suggests the strain’s ability to produce simple aromatic polyketides like chalcones and resorcinols, which often serve as precursors for complex natural products or exhibit independent bioactivities [81].

The genome annotation reveals the presence of several NRPSs with a range of bioactive properties. Notable examples with high similarity ≥ 50% include alpiniamide and valinomycin/montanastatin, all of which have significant potential [82,83,84]. Other compounds like griseobactin, minimycin, and heat-stable antifungal factor further demonstrate the strain’s ability to produce a variety of peptides with diverse bioactivities, including antibacterial, antifungal, and anticancer activities. Beyond PKS, NRPS, and terpenoid clusters, the genome of *Streptomyces* sp. ACT158 includes BGCs for RiPPs and siderophores, which contribute to its metabolic versatility and potential bioactivity both antimicrobial and antitumor potential [85].

## 5. Conclusions

Comparative genomic analysis placed ACT158 within the *Streptomyces cavourensis* species complex, with ANI and DDH values supporting its taxonomic assignment. Despite its close relationship to other *S. cavourensis* strains, ACT158 harbors unique gene families, likely contributing to its specialized ecological adaptations and secondary metabolite production capabilities. These unique genetic features emphasize the significance of exploring under-represented environments like wetlands for discovering actinomycetes with novel biosynthetic traits. Overall, *Streptomyces* sp. ACT158 exemplifies the potential of actinomycetes from unique ecological niches to address critical challenges in medicine and industry. Its phenotypic adaptability, metabolic diversity, and genomic insights indicate its suitability for developing novel bioactive compounds. Future research should prioritize isolating and characterizing the specific secondary metabolites produced by ACT158 to unlock its full biotechnological potential. In addition, optimizing fermentation conditions to enhance metabolite yield and exploring its environmental applications could pave the way for sustainable exploitation of this promising strain.

## Figures and Tables

**Figure 1 microorganisms-13-00576-f001:**
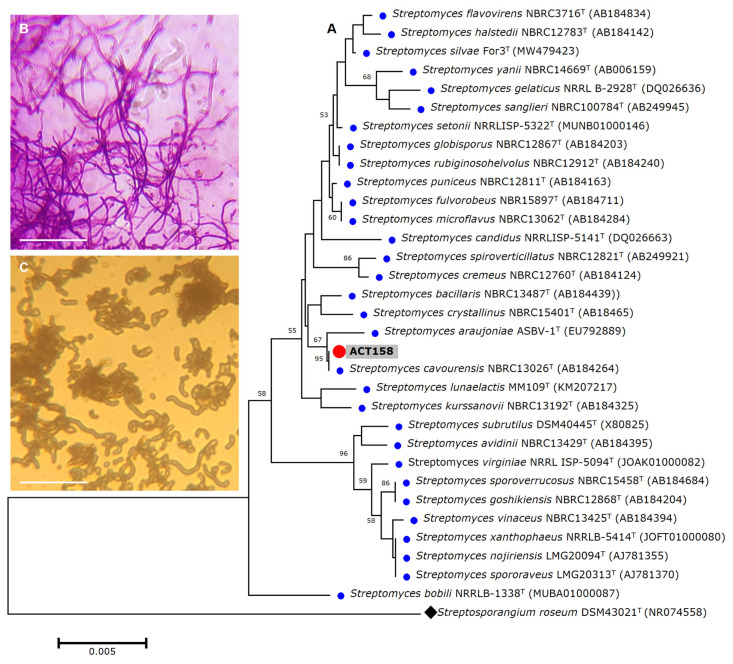
The phylogenetic relationships based on 16S rRNA gene sequences between *Streptomyces* sp. ACT158 and other related taxa generated by neighbor-joining. Bootstrap values (expressed as percentages of 1000 replications) greater than 50% are shown. Bar, 0.005 substitutions per nucleotide position (**A**). Gram stain (**B**), and cell morphology (**C**) of the ACT158 strain. The ACT158 strain was grown on SFM medium and the aerial mycelia morphology was viewed using optical microscopy (×100).

**Figure 2 microorganisms-13-00576-f002:**
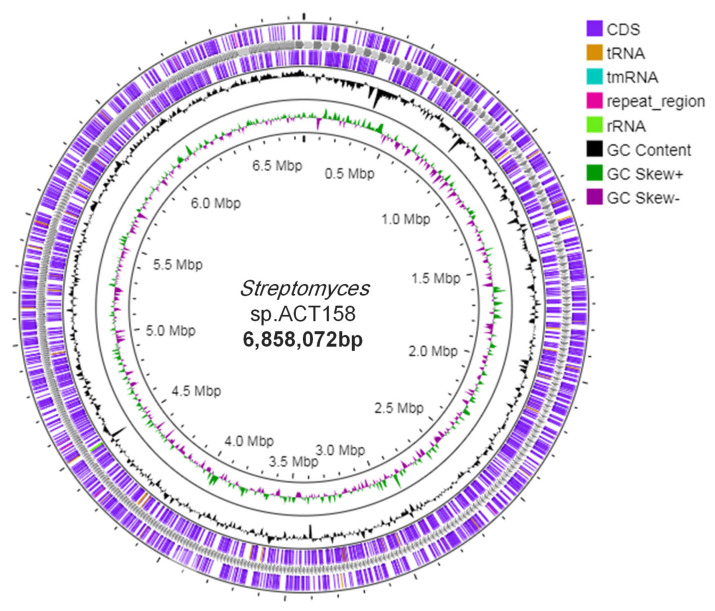
Circular chromosome map of the whole genome of *Streptomyces* sp. ACT158 generated through Proksee (https://proksee.ca, accessed on 10 December 2024).

**Figure 3 microorganisms-13-00576-f003:**
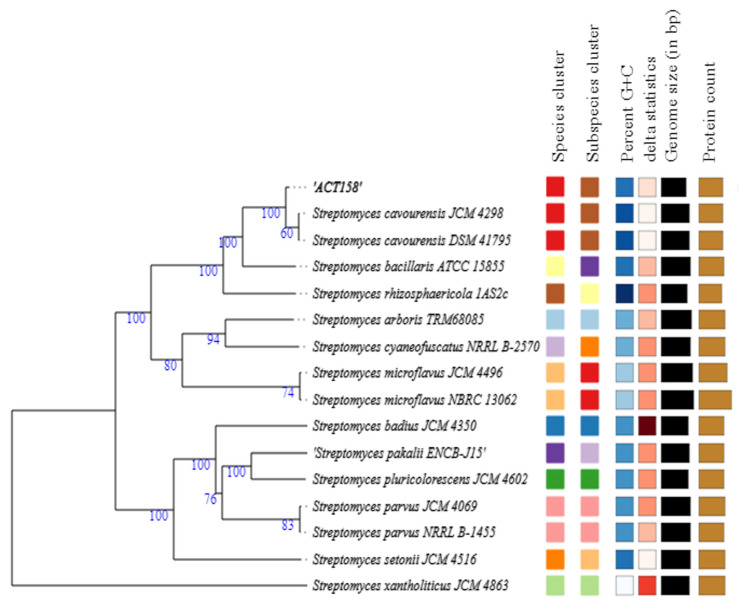
Phylogenomic tree based on genome sequences of ACT158, and other related taxa inferred with FastME 2.1.6.1 from GBDP distances. The branch lengths are scaled in terms of GBDP distance formula d5. The numbers above branches are GBDP pseudo-bootstrap support values > 60% from 100 replications, with an average branch support of 89.1%. The tree was rooted at the midpoint.

**Figure 4 microorganisms-13-00576-f004:**
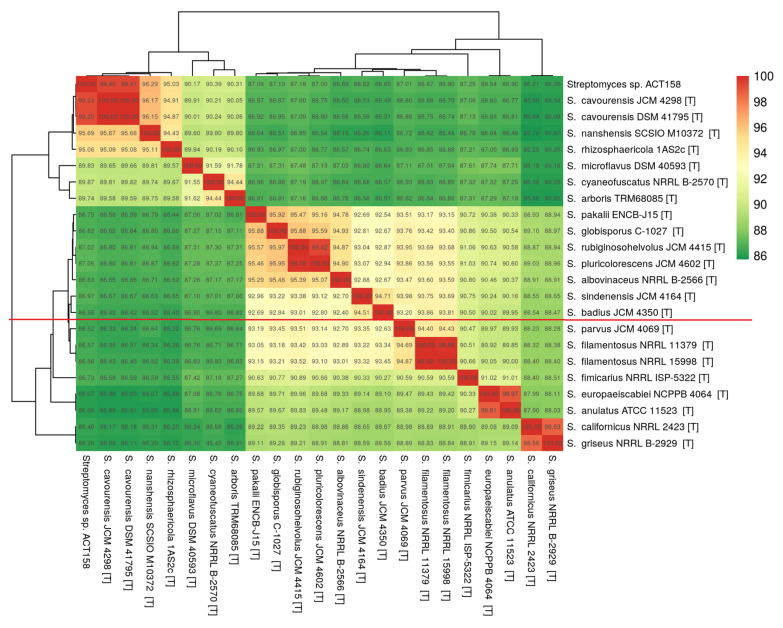
Digital DNA-DNA hybridization (dDDH) of ACT158, and other related taxa calculated using the Genome-to-Genome Distance Calculator 2.1 on the DSMZ website.

**Figure 5 microorganisms-13-00576-f005:**
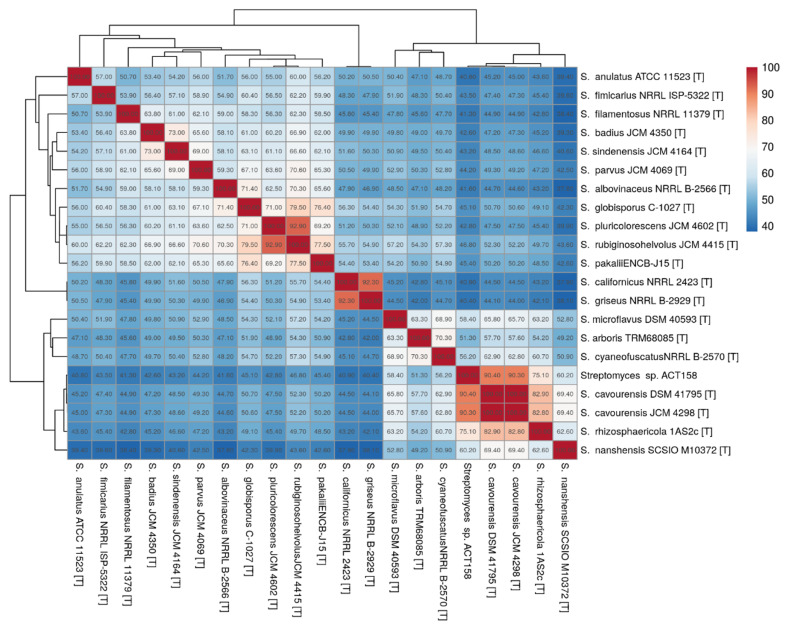
Average nucleotide identity (ANI) calculated using the BLAST-based OrthoANI heatmaps of ACT158 and other closely related Streptomyces species.

**Figure 6 microorganisms-13-00576-f006:**
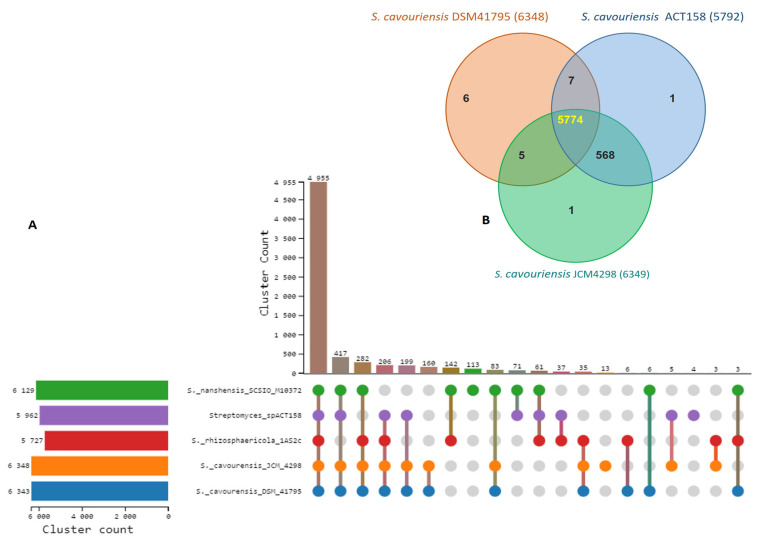
Ortholog clusters analysis between *Streptomyces* sp. ACT158 and their closely related *Streptomyces* species. (**A**) Ortholog clusters count, (**B**) Venn diagram represents distribution of shared and unique gene clusters among ACT158, *S. cavourensis* DSM41795, and *S. cavourensis* JCM4298.

**Figure 7 microorganisms-13-00576-f007:**
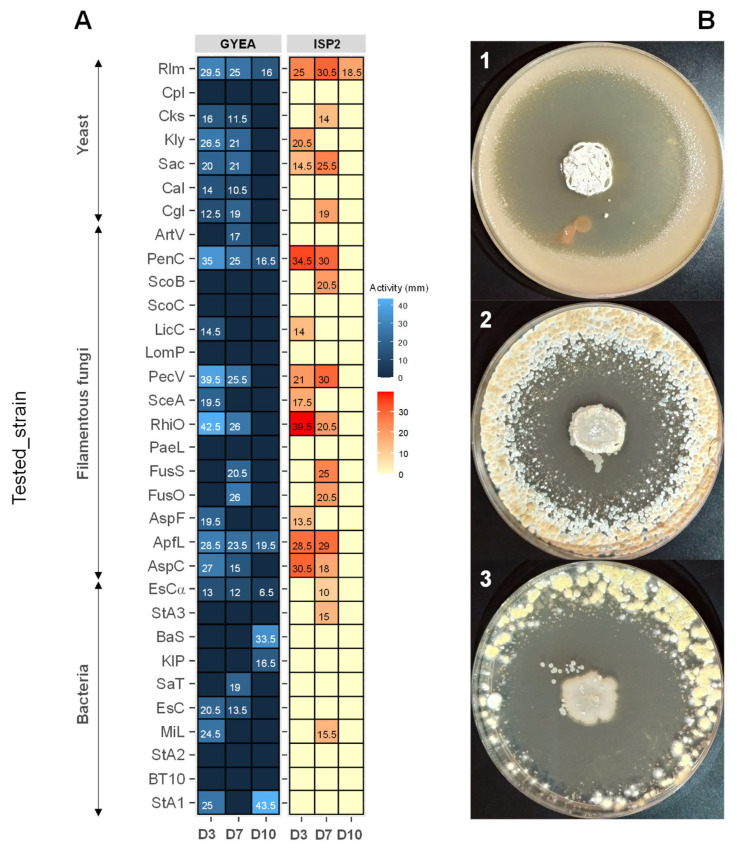
Antimicrobial activity (mm) of *Streptomyces* sp. ACT158 on GYEA and ISP2 after 3, 7, and 10 days of incubation (**A**). Antifungal activity of *Streptomyces* sp. ACT158 on the growth of different test organisms (**B**). The active ACT158 strain was inoculated as a spot in the center of ISP2 plates at 30 °C for 7 days. After, the plates were then covered with 10 mL of GYEA previously inoculated with target fungi (**1**: *Rhodotorula mucilaginosa*, **2**: *Penicillium chrysogenum*, **3**: *Rhizopus oryzae*).

**Figure 8 microorganisms-13-00576-f008:**
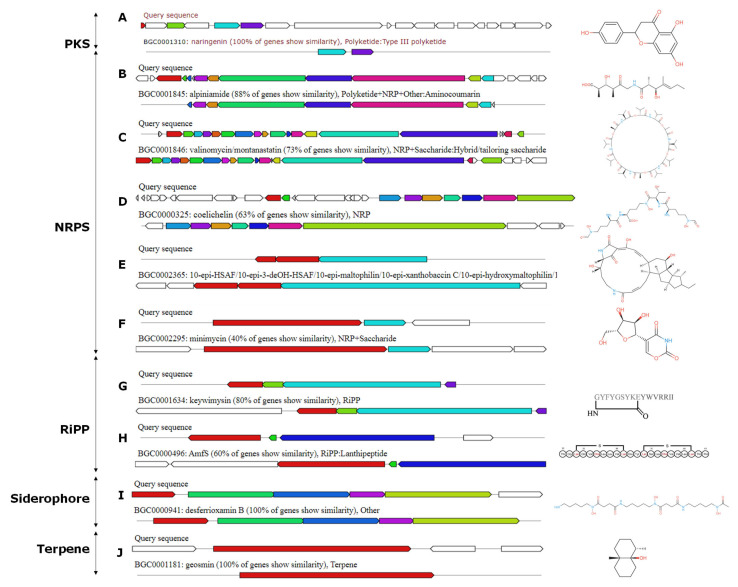
AntiSMASH predicted biosynthetic gene clusters and their predicted core structures from *Streptomyces* sp. ACT158 genome.

**Table 1 microorganisms-13-00576-t001:** General features of *Streptomyces* sp. ACT158 draft genome sequence.

Characteristics	Values
Total Length (bp)	6,858,072
Number of coding sequences	6831
GC Content (%)	71.8%
N50	8070
Gap Ratio (%)	0.0%
No. of CDSs	5122
No. of rRNA	3
No. of tRNA	54
No. of CRISPRS	8
Coding Ratio (%)	70.1%

**Table 2 microorganisms-13-00576-t002:** Putative secondary metabolites gene clusters in *Streptomyces* sp. ACT158 predicted by antiSMASH.

Type	MIBiG Accession	Known Cluster Blast (Biosynthetic Gene)	Similarity	Potential Activity
NRPS-T1PKS	BGC0001310	naringenin	100%	Antifungal, antiviral
NRPS-T1PKS	BGC0002591	aurachin C	20%	Antibacterial
NRPS-T1PKS	BGC0000236	kinamycin	13%	Antitumor
NRPS-T1PKS	BGC0000273	steffimycin D	16%	Antitumor
NRPS-T1PKS	BGC0000028	bafilomycin B1	27%	Antibacterial, antitumor
NRPS-T1PKS	BGC0001477	lydicamycin	32%	Antibacterial
NRPS-T1PKS	BGC0001348	JBIR-100	38%	Antimicrobial
NRPS-T1PKS	BGC0001700	niphimycins C-E	29%	Antimicrobial
NRPS-T2PKS	BGC0000234	jadomycin	42%	Antimicrobial
NRPS-T3PKS	BGC0000282	alkylresorcinol	100%	Antibacterial
NRPS	BGC0002001	crochelin	16%	Antimicrobial, antitumor
NRPS	BGC0001845	alpiniamide	88%	Antimicrobial
NRPS	BGC0000453	valinomycin/montanastatin	73%	Antimicrobial
NRPS	BGC0000325	coelichelin	63%	/
NRPS	BGC0001368	JBIR-126	7%	Antimicrobial
NRPS	BGC0000368	griseobactin	30%	Antibacterial
NRPS	BGC0002441	o-dialkylbenzene 1	12%	/
NRPS	BGC0002295	minimycin	40%	Antimicrobial
NRPS	BGC0002365	heat-stable antifungal factor	50%	Antifungal
NRPS	BGC0000375	Indigoidine	40%	Antibacterial, antioxidant
RiPP	BGC0000583	Linaridin	20%	Antibacterial
RiPP	BGC0000496	AmfS	60%	Antimicrobial
RiPP	BGC0000606	lactazole	22%	Antimicrobial, antitumor
RiPP	BGC0001634	Keywimysin	80%	Antibacterial
Siderophore	BGC0000941	desferrioxamin B	100%	Antimicrobial, Iron Chelation
butyrolactone	BGC0001778	showdomycin	29%	Antimicrobial
butyrolactone	BGC0000038	coelimycin	8%	Antibacterial
Ectoine	BGC0000858	ectoine	75%	Osmolyte
Terpene	BGC0001181	Geosmine	100%	/
Terpene	BGC0000664	Isorenieratene	42%	/
Terpene	BGC0000663	hopene	46%	

## Data Availability

The original contributions presented in this study are included in the article/Appendix A. Further inquiries can be directed to the corresponding author.

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
