# Peer review of "Genomic Insights and Antimicrobial Potential of Newly Streptomyces cavourensis Isolated from a Ramsar Wetland Ecosystem"

_microorganisms, 2025, doi:10.3390/microorganisms13030576_

Round 1
Reviewer 1 Report
Comments and Suggestions for Authors
In this work by Mabrouka Benhadj et al. the authors isolated from a RAMSAR wetland ecosystem and characterized Streptomyces cavourensis strain ACT158 using its morphology, phylogenetic analysis and modern methods AAI, ANI, dDDH. Comparative genomic analysis showed high genomic synteny with closely related Streptomyces strains. Nevertheless, Streptomyces sp. CT158 have unique gene clusters. It was shown antimicrobial and antifungal potential of Streptomyces sp. ACT158.
Comments and suggestions to the authors:
The caption to Figure 1 does not contain a reference to photos B and C and there is no link in the manuscript to these photos. This information should added.
The authors assumed that the Streptomyces sp. ACT158 might have antitumor activity. It will be rather interesting to investigate anticancer activity of Streptomyces sp. ACT158 against HeLa cell lines f.e. like in Madheslu et al. 2018 “Taxonomic Characterization and Antagonistic Efficacy of Streptomyces cavourensis SKCMM1 Isolated from Sediment of Pichavaram Mangrove Forest”. Did the authors consider such an experiment?
Author Response
In this work by Mabrouka Benhadj et al. the authors isolated from a RAMSAR wetland ecosystem and characterized Streptomyces cavourensis strain ACT158 using its morphology, phylogenetic analysis and modern methods AAI, ANI, dDDH. Comparative genomic analysis showed high genomic synteny with closely related Streptomyces strains. Nevertheless, Streptomyces sp. CT158 have unique gene clusters. It was shown antimicrobial and antifungal potential of Streptomyces sp. ACT158.
Comments and suggestions to the authors:
- The caption to Figure 1 does not contain a reference to photos B and C and there is no link in the manuscript to these photos. This information should add.
Response
Thank you for your valuable feedback and suggestions. We have addressed the issue regarding the caption of Figure 1 by adding the necessary references to photos B and C, and ensuring the manuscript links appropriately to these images.
- The authors assumed that the Streptomyces sp. ACT158 might have antitumor activity. It will be rather interesting to investigate anticancer activity of Streptomyces sp. ACT158 against HeLa cell lines f.e. like in Madheslu et al. 2018 “Taxonomic Characterization and Antagonistic Efficacy of Streptomyces cavourensis SKCMM1 Isolated from Sediment of Pichavaram Mangrove Forest”. Did the authors consider such an experiment?
Response
We appreciate your suggestion to explore the anticancer activity of Streptomyces sp. ACT158 against HeLa cell lines. While this is an interesting and important direction for future research, our current study focused on antimicrobial and antifungal potential. We have noted your reference to Madheslu et al. (2018) and agree that incorporating anticancer activity could add significant value to understanding the full spectrum of biological activity of Streptomyces sp. ACT158. We will consider this experiment as a potential follow-up study in future research, and we will include this as a suggestion for further investigation in the conclusion of our manuscript.
Reviewer 2 Report
Comments and Suggestions for Authors
Dear authors, the manuscript titled "Genomic Insights and Antimicrobial Potential of Newly Isolated Streptomyces cavourensis from a RAMSAR Wetland Ecosystem" presents significant findings with substantial applicative value. The study effectively addresses the pressing issue of antimicrobial resistance by identifying and characterizing a promising Streptomyces strain with broad-spectrum bioactivity. The comprehensive genomic analysis, including whole-genome sequencing, biosynthetic gene cluster identification, and comparative genomics, provides valuable insights into the strain’s potential for novel bioactive compound production.
The manuscript is well-structured, with a clear and logical presentation of methods, results, and their implications. The depth of genomic analysis, combined with its ecological and biotechnological relevance, makes this study a strong contribution to the field of microbial biotechnology and natural product discovery. The writing is of high quality, and the article is well-suited for publication following minor editorial revisions. The comments that could be considered by authors for improve a publication are listed below.
1. The quality of figures, resolution could be immproved
2. Line 550. Authors could consider adding a 'Conclusion' header in this section
Author Response
Dear authors, the manuscript titled "Genomic Insights and Antimicrobial Potential of Newly Isolated Streptomyces cavourensis from a RAMSAR Wetland Ecosystem" presents significant findings with substantial applicative value. The study effectively addresses the pressing issue of antimicrobial resistance by identifying and characterizing a promising Streptomyces strain with broad-spectrum bioactivity. The comprehensive genomic analysis, including whole-genome sequencing, biosynthetic gene cluster identification, and comparative genomics, provides valuable insights into the strain’s potential for novel bioactive compound production.
The manuscript is well-structured, with a clear and logical presentation of methods, results, and their implications. The depth of genomic analysis, combined with its ecological and biotechnological relevance, makes this study a strong contribution to the field of microbial biotechnology and natural product discovery. The writing is of high quality, and the article is well-suited for publication following minor editorial revisions. The comments that could be considered by authors for improve a publication are listed below.
- The quality of figures, resolution could be improved
Response
Thank you very much for your positive and constructive feedback. We are grateful for your appreciation of the study’s findings. We acknowledge the suggestion and the figures were enhanced in revised manuscript to ensure that they meet the required standards for clarity and readability.
- Line 550. Authors could consider adding a 'Conclusion' header in this section
Response
We agree with your suggestion. We have incorporated this change in the revised manuscript to provide a clearer summary of the key findings.